# ALIGNING TEACHER WITH STUDENT PREFERENCES FOR TAILORED INSTRUCTION TUNING DATASET GENERATION

## ABSTRACT

Enhancing the reasoning abilities of lightweight language models (LMs) for tasks like decision-making often relies on instruction-tuning, a method that trains LMs to mimic the reasoning process using labeled question-rationale pairs, known as instruction-tuning datasets, which are typically generated by more powerful teacher LMs. However, current methods for generating these instruction-tuning datasets tend to focus solely on the quality of the questions and rationales from the teacher model's perspective, often neglecting the learning preferences of the student language model. To fill this gap, we propose **ARTE** (**A**ligning Teache**R** with Studen**T** Preferenc**E**s), a novel framework that adapts the teacher LM's outputs to the student's preferences, inspired by "responsive teaching" in pedagogy. Our method involves three key steps: (1) generating draft question-rationale pairs from the teacher model, (2) collecting the student's preferences on these draft pairs via one-shot in-context learning, and (3) aligning the teacher model using Direct Preference Optimization (DPO), then finally curating tailored question-rationale pairs from the aligned teacher for student training. Through extensive experiments on academic reasoning benchmarks, we demonstrate that student models fine-tuned with tailored datasets by ARTE achieve significant improvements across various reasoning tasks, outperforming existing instruction-tuning datasets. Moreover, we thoroughly investigate the generalization of ARTE, including the generalization of fine-tuned student models in reasoning ability and the generalization of aligned teacher models to generate tailored training data across tasks and students.

## 1 INTRODUCTION

Lightweight language models (LM) have achieved great success in various tasks like writing and chatting, while it is still challenging for them to perform well in reasoning tasks which is crucial for many applications like decision making and problem solving (Hug, 2024; Banks & Warkentin, 2024; Hu et al., 2024; AI, 2024; Muralidharan et al., 2024). Instruction Tuning is a popular method for improving the reasoning ability of lightweight language models by imitating the reasoning processes on labeled question-rationale pairs, namely the instruction-tuning dataset, with Supervised Fine-tuning (Mitra et al., 2023; Hsieh et al., 2023; Xu et al., 2023; Luo et al., 2023; Mukherjee et al., 2023). Given the high cost and scalability issues of human-labeled instruction-tuning datasets, it is common to leverage the Knowledge Distillation (KD) paradigm (Hinton et al., 2015) to distill an instruction-tuning dataset from a powerful teacher language model. Students are then trained on these distilled question-rationale pairs to improve their reasoning ability with Supervised Fine-tuning (Peng et al., 2023; Ivison et al., 2023; Mukherjee et al., 2023; Xu et al., 2023).

To better distill instruction-tuning datasets from a teacher model, the quality of the generated question-rationale pairs is crucial. For questions, Taori et al. (2023); Wang et al. (2022) proposed to use ROUGE-L (Lin, 2004) to filter out the repeated questions to improve the diversity of the generated questions. Xu et al. (2023) propose Evo-Instruct to rewrite the seed questions step by step into more complex questions to improve the complexity of the generated questions. As for rationales, Hsieh et al. (2023); Mukherjee et al. (2023) leverage chain-of-thought (Wei et al., 2022; Yao et al., 2022) to guide the teacher model in producing rationales with detailed reasoning processes to improve the quality of the generated rationales. Mitra et al. (2023) contends that the reasoning pro-

cesses in the rationales should be adaptive to different question types. While these methods improve question and rationale quality from the teacher model's perspective, they overlook the learning preferences of the student model. In practice, an ideal teacher model should generate question-rationale pairs that are tailored to the student's preferences and learning needs, to lower the learning barrier, akin to the concept of 'responsive teaching' in pedagogy (Gay, 2018; Hattie & Timperley, 2007).

To fill this gap, we propose **ARTE**, short for **A**ligning Teache**R** with Studen**T** Preferenc**E**s, a novel framework that aligns the teacher language model with the student model's preferences to generate tailored instruction-tuning datasets. Our framework is illustrated in Figure 1. There are three main steps in ARTE: 1) *Knowledge Elicitation* (Section 2.1): We prompt the teacher model with seed questions to generate draft question-rationale pairs. 2) *Preference Collection* (Section 2.2): We collect the student's preferences for each draft pair using one-shot in-context learning on a sampled subset of the validation set of the target reasoning task. 3) *Preference Alignment* (Section 2.3): We align the teacher model with the student's preferences using Direct Preference Optimization (DPO) (Rafailov et al., 2023) to improve instructional content. Finally, we repeat the first step with the aligned teacher model to curate tailored instruction-tuning datasets, which are used to fine-tune the student model via supervised learning.

Extensive experiments on academic reasoning benchmarks demonstrate that ARTE outperforms existing instruction-tuning datasets by a significant margin (Peng et al., 2023; Ivison et al., 2023; Mukherjee et al., 2023; Xu et al., 2023; Ding et al., 2023). Specifically, our method achieves improvements of $9.6\%$, $1.0\%$, $0.8\%$, and $8.5\%$ on logic reasoning, commonsense reasoning, math reasoning, and knowledge reasoning tasks, respectively, in the Big-Bench-Hard (Suzgun et al., 2023) benchmark. We also show that ARTE generalizes well across different reasoning tasks and student models. 1) We investigate generalization in reasoning tasks through in-context learning on out-of-domain benchmarks, including PIQA (Bisk et al., 2020), CommonsenseQA (Talmor et al., 2019), ARC-Easy (Clark et al., 2018), ARC-Challenge (Clark et al., 2018), GSM8K (Cobbe et al., 2021), and MMLU (Hendrycks et al., 2021). In these benchmarks, ARTE consistently outperforms baseline methods, validating the effectiveness of aligning the teacher model with student preferences. 2) We further explore the generalization of the aligned teacher model across different tasks and student models. Results indicate that a teacher model aligned with BBH can generate high-quality examples for unseen reasoning tasks, such as ARC or GSM8K. Furthermore, a teacher model aligned with a specific student model, such as Gemma-2B, can generate tailored training examples for other models with similar parameter capacities, such as Qwen1.5-1.8B or CodeGemma-2B.

To summarize, our contributions are as follows:

- We propose ARTE, a novel framework inspired by responsive teaching that aligns the teacher model with the student model's preferences to generate tailored instruction-tuning datasets for reasoning tasks.

- Extensive experiments on in-domain and out-of-domain reasoning benchmarks demonstrate that the student model fine-tuned with in-domain datasets generated through ARTE outperforms models trained on existing instruction-tuning datasets.

- We also explore the generalization capability of the aligned teacher model, showing that it can generate high-quality training examples across different reasoning tasks and for other student models with similar capacities.

## 2 ARTE

We introduce ARTE, dubbed **A**ligning Teache**R** with Studen**T** Preferenc**E**s, a novel framework that aligns the teacher $LM_t$ with student $LM_s$'s preferences to generate tailored question-rationale pairs for the target task $T$. Our overall framework is illustrated in Figure 1. There are three main steps in our framework: 1) *Knowledge Elicitation*: we prompt the teacher model with seed questions to generate a draft dataset consisting of draft questions and corresponding draft rationales. 2) *Preference Collection*: We collect the preference scores from the student model for both draft questions and rationales using the one-shot in-context learning performance of the student model on the preference set as a proxy. 3) *Preference Alignment*: Based on the preference score of questions and rationales, we align the teacher model with the student model's preferences through Direct Preference Optimization (DPO). Finally, we repeat the first step with the aligned teacher model to curate tailored

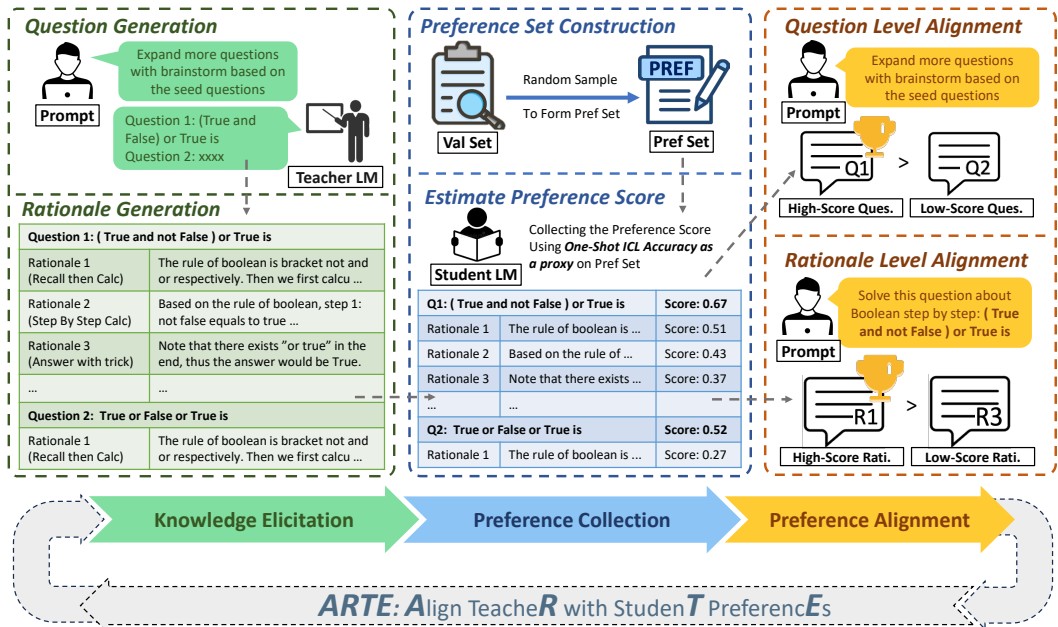

Figure 1: The overall framework of ARTE.

training pairs for the student model and use them to Supervised Fine-Tune the student model. In the following, we will elaborate on each step in detail.

## 2.1 KNOWLEDGE ELICITATION

In this step, we construct a dataset $D_{\text{draft}} = \{(q, R_q)\}$ for the target task $T$, where $q$ is the draft question and $R_q = \{r_q\}$ is the set of rationales associated with the question $q_i$. For question generation, we prompt the teacher model $LM_t$ with seed questions to expand them into $m$ draft questions. Specifically, we use three seed questions to construct one question generation prompt which guides the teacher model to brainstorm multiple questions. We decode the question generation prompt at a temperature $1.0$ multiple times until we get $m$ draft questions.

For rationale generation, inspired by the observation that one question may have multiple different reasoning processes leading to the same answer and the optimal reasoning process may vary for different questions and different models (Mitra et al., 2023). Then, for each draft question $q$, $LM_t$ is guided to generate $n$ rationales $r_q$ to form a rationale set $R_q$ using different reasoning techniques, such as *Explain Like I'm Five*, *Step-by-Step*, or *Math Symbols*. Finally, we combine each draft question $q$ with its corresponding rationales $R_q$ to form a draft sample $(q, R_q)$. Since each draft question $q$ would be expanded into $n$ rationales, there are $m \times n$ draft pairs $E_{\text{draft}} = \{(q, r_q)\}$ in the draft dataset $D_{\text{draft}}$.

## 2.2 PREFERENCE COLLECTION

In this step, we aim to collect the preferences of the student model for question-rationale pairs during Supervised Fine-Tuning (SFT). To achieve this, a reliable metric that can reflect the student model's preference for draft pairs is needed. Recent research (Dai et al., 2023; Von Oswald et al., 2023) suggests that language models perform gradient-based updates during In-Context Learning (ICL), similar to the process of SFT. Additionally, Lin et al. (2023) demonstrates that ICL can effectively replace SFT in alignment tasks. All these findings suggest that the one-shot ICL performance of the student model with draft pairs can be used as a proxy for preference scores during SFT. To further empirically validate this, we first sample 50 questions from the validation set to construct the preference set $D_{\text{pref}}$. Then, we collect the one-shot ICL performance of the student model with different draft pairs on the preference set $D_{\text{pref}}$.

Finally, we trained the student model (Gemma-2B) using pairs from the top-25% (highest ICL performance), bottom-25% (lowest ICL performance), and a random 25% sample from the full draft example dataset and evaluated them on the Big-Bench-Hard (BBH) test set. The results in Table 1 show that models trained with higher ICL performance pairs significantly outperform those trained with pairs with lower ICL performance, further confirming the reliability of using ICL as a proxy for preference.

|  | **Bot-25%** | **Top-25%** | **Rand-25%** |
|---|---|---|---|
| **Acc**. | 35.57% | 42.78% | 39.94% |

Table 1: Accuracy of Gemma-2B trained with different data subsets (Bottom-25%, Top-25%, Random-25%) based on ICL performance, evaluated on BBH test set.

With the above findings, we define the preference score of a rationale $r$ and a question $q$ as the average one-shot ICL performance of the student model on the preference set $D_{\text{pref}}$. Formally, for rationale $r$ associated with question $q$, the preference score $s_r$ is defined as:

$$s_r = \frac{1}{|D_{\text{pref}}|} \sum_{(q',a') \in D_{val}} \mathbb{I}(LM_s(q,r;q') = a') \tag{1}$$

where $LM_s(q,r;q')$ denotes the student model's answer to question $q'$ when $q$ and $r$ are used as the one-shot in-context example, and $\mathbb{I}(\cdot)$ is the indicator function. The preference score $s_q$ of question $q$ is defined as the average of the preference scores of its associated rationales:

$$s_q = \frac{1}{|R_q|} \sum_{r \in R_q} s_r \tag{2}$$

where $R_q$ is the set of rationales associated with question $q$.

## 2.3 PREFERENCE ALIGNMENT

After collecting the preference score of questions and rationales, we aim to align the teacher model with the student model's preferences to generate tailored pairs for the target task $T$. There are two main aspects to align, *Question Generation* and *Rationale Generation*, towards generating not only tailored rationales but also tailored questions. Direct Preference Optimization (DPO) (Rafailov et al., 2023) is selected to align the teacher model with the student model's preferences due to its stability and simplicity compared to other methods like PPO (Schulman et al., 2017). For question level alignment, for the input question generation prompt, we randomly sample out $k$ draft questions with the top-25% and bottom-25% preference scores as the chosen and rejected responses respectively. For rationale level alignment, for each question $q$, we select the rationale from $R_q$ with the highest preference score as the chosen response and the rationale with the lowest preference score as the rejected response. Note that the input prompt in rationale level alignment is just a naive step-by-step prompt as shown in Table 14. Finally, we mix the question level and rationale level DPO datasets to perform DPO to align the teacher model with the student model's preferences.

After the alignment, we repeat the first step to generate tailored pairs for the target task $T$ and use them to Supervised Fine-Tune the student model. Note that when generating tailored questions and rationales, we use the same input prompt from the alignment step and decode at temperature $\tau = 0$ for deterministic decoding. More details can be found in Appendix E.

## 3 EXPERIMENTS

In this section, we conduct experiments to evaluate the effectiveness of our method to transfer the reasoning capabilities from the teacher model to the student model to target tasks Big-Bench-Hard (Suzgun et al., 2023).

## 3.1 EXPERIMENTAL SETUP

**Datasets**  We conduct experiments on the Big-Bench-Hard (Suzgun et al., 2023), a suite of 23 challenging tasks from the Big-Bench benchmark (Srivastava et al., 2022). The Big-Bench-Hard is designed to test the generalization ability of models on a wide range of tasks, including logical reasoning, commonsense reasoning, world knowledge, math ability, etc.

For better presentation, we divide the tasks into four categories based on the capabilities required: (1) *logical reasoning*, (2) *commonsense reasoning*, (3) *math reasoning* and (4) *world knowledge*, denoted as *BBH-Logic*, *BBH-CS*, *BBH-Math* and *BBH-Knowl* respectively. Details of the task categories are shown in Appendix D. Table 2 shows the statistical information of the tasks in Big-Bench-Hard. For each task, we sample 40 questions from the original Big-Bench dataset to form the preference set.

| Category | #Tasks | #Pref | #Test |
|----------|--------|-------|-------|
| BBH-Logic | 10 | 400 | 3,146 |
| BBH-CS | 7 | 280 | 1,615 |
| BBH-Math | 3 | 120 | 750 |
| BBH-Knowl | 3 | 120 | 750 |
| Total | 23 | 920 | 6,261 |

Table 2: Statistics of the Big-Bench-Hard dataset.

**Baselines** Since We compare our method with the following instruction-tuning datasets as academic baselines: 1) GPT-4-LLM (Peng et al., 2023) is an instruction-tuning dataset distilled from `GPT-4` with Self-Instruct (Wang et al., 2022). 2) Tulu-v2 (Ivison et al., 2023) is a mix of multiple existing high-quality instruction-tuning datasets, including FLAN (Longpre et al., 2023), OpenAssistant (Köpf et al., 2024), etc. 3) OpenOrca is a reproduction of Orca (Mukherjee et al., 2023), which augments FLAN data with additional `GPT-4` or `GPT-3.5-Turbo` generated explanations. 4) WizardLM-Evol-Instruct (Xu et al., 2023) is instruction-fine-tuning dataset which is distilled from the `GPT-4` through *Evol-Instruct*. 5) UltraChat (Ding et al., 2023) contains high-quality multi-turn dialogues and covers a wide range of topics and instructions.

Apart from the above academic baselines, we also compare our method with ablation baselines to testify the effectiveness of the two different alignment aspects in our method: 1) *Original Teacher*: the dataset constructed by the draft questions and its corresponding rationale with the highest preference score in the preference set. Namely, in this baseline, both the questions and rationales are generated by the original teacher model. 2) *Rationale Only*: In this setting, only the rationales are generated by the aligned teacher model, while the questions are generated by the original teacher model. 3) *Question Only*: In this setting, only the questions are generated by the aligned teacher model, while the rationales are generated by the original teacher model. To ensure a fair comparison, we sample out $6,750$ examples from each baseline dataset to conduct the experiment. Supervised Fine-Tune is used to train the student model on these datasets with 1 epoch using the Adam optimizer (Kingma & Ba, 2014) and a learning rate of $2e-5$. We use 3% of the total training steps for cosine learning rate warm-up and adopt a linear decay learning rate schedule.

**Implementation** In our experiments, we adopt Llama-3-70B-Instruct (Meta, 2024) as the teacher model and Gemma-2B (Banks & Warkentin, 2024) as the student model. We run all of our experiments with 5 different seeds and report the mean of the results in the tables below.

In the *Knowledge Elicitation* step, 1) *Question Generation*: We decode the question generation prompt in Table 15 with the aligned teacher at a temperature of $1.0$ until we achieve a total of 250 draft questions per task. 2) *Rationale Generation*: For each draft question, the teacher model is prompted to generate $n = 4$ rationales using the prompt in Table 16 with diverse reasoning techniques. Each draft question with its corresponding rationales forms a draft sample $(q, R_q)$.

In the *Preference Collection* step, we collect the preference scores of questions and rationales through in-context learning on the preference set. Specifically, we use the prompt template in Table 17 and decoding at a temperature $\tau = 0$ to ensure the model's answer is deterministic.

In the *Preference Alignment* step, we utilize preference scores to create the DPO dataset and align the teacher model with the student model's preferences. For question generation, we select 50 draft questions with the highest and lowest preference scores per task as preferred and dispreferred respectively, creating 50 DPO training examples per task at the question level. For rationale generation, the highest and lowest score rationales per question are selected similarly, resulting in 250 DPO training examples at the rationale level per task. These datasets are combined to form the final DPO dataset. We employ Direct Preference Optimization (DPO) to align the models at a learning rate of $1e-7$, batch size of 16, for 1 epoch, incorporating a linear warm-up over 10% of training steps followed by a cosine decay learning rate schedule.

| Models | BBH-Logic | BBH-CS | BBH-Math | BBH-Knowl | BBH |
|---|---|---|---|---|---|
| *zero-shot result* | | | | | |
| Vanilla Gemma-2B | 0.80 | 0.00 | 6.00 | 0.00 | 1.09 |
| + GPT-4-LLM | 1.35 | 0.00 | 1.60 | 0.93 | 1.00 |
| + Tulu-v2 | 6.15 | 5.83 | 16.00 | 0.67 | 6.33 |
| + UltraChat | 3.74 | 0.56 | 7.07 | 8.00 | 3.82 |
| + WizardLM | 6.24 | 2.14 | 12.27 | 15.33 | 6.88 |
| + OpenOrca | 5.17 | 4.68 | 20.13 | 8.53 | 7.42 |
| + Original Teacher | 35.57 | 40.61 | 33.87 | 40.27 | 37.55 |
| + Rationale Only | 36.07 | 37.54 | 37.87 | 40.53 | 37.94 |
| + Question Only | **40.29** | 43.45 | 40.80 | 40.67 | 41.73 |
| + ARTE | 39.07 | **44.44** | **42.00** | **42.53** | **41.96** |
| *three-shot result* | | | | | |
| Vanilla Gemma-2B | 28.16 | 40.92 | 42.13 | 44.40 | 35.57 |
| + GPT-4-LLM | 29.58 | 40.79 | 36.27 | 49.20 | 36.11 |
| + Tulu-v2 | 30.28 | 39.74 | 38.67 | 50.00 | 36.35 |
| + UltraChat | 31.01 | 41.05 | **47.73** | 47.07 | 37.28 |
| + WizardLM | 29.62 | 43.44 | 45.33 | 45.33 | 37.29 |
| + OpenOrca | 28.91 | 41.91 | 43.47 | 49.87 | 36.92 |
| + Original Teacher | 34.32 | 43.06 | 44.13 | 52.13 | 40.04 |
| + Rationale Only | 37.43 | 42.17 | 47.47 | 49.73 | 41.30 |
| + Question Only | **40.39** | 43.85 | 41.73 | 51.87 | 42.78 |
| + ARTE | 39.86 | **44.45** | 46.13 | **53.87** | **43.44** |

Table 3: Accuracy (%) of the student model Gemma-2B fine-tuned with different instruct-tuning datasets on Big-Bench-Hard under zero-shot and three-shot settings. Cells are colored blue if the method improves over Vanilla Gemma-2B, and orange if it declines.

Finally, we repeat the first step to generate tailored examples for BBH tasks and use them to Supervised Fine-Tune the student model with the same hyperparameters as baselines. All the experiments are conducted on 8 * NVIDIA A100 80G GPUs, which will cost about 1 day in total.

## 3.2 RESULTS

Table 3 shows the results of our method and baselines on the Big-Bench-Hard dataset under the zero-shot and three-shot in-context learning settings. 1) Compared to academic baselines: we can see that our method outperforms all the baselines including the baselines that simply imitate the reasoning process of the most powerful LLM (GPT-4-LLM), mix multiple existing high-quality instruction-tuning datasets (Tulu-v2), distill the teacher model with hand-crafted curations (OpenOrca and WizardLM), and high-quality multi-turn dialogues (UltraChat). 2) Compared to the ablation baselines: our method outperforms both question-only and rationale-only baselines by a large margin, indicating that both the questions and rationales generated by the aligned teacher model are important for the student model to achieve better performance. Besides, notice that the performance of the question-only baseline achieves better performance than the rationale-only baseline. This interesting phenomenon indicates that in Knowledge Distillation for the language model, the quality of questions plays a more important role than the quality of rationales.

## 4 ANALYSIS ON GENERALIZATION

Although ARTE outperforms the baselines on the Big-Bench-Hard dataset, the generalization performance of it is still unclear. Thus in this section, we present a thorough examination of its generalizability in the following two aspects: 1) Generalization of the fine-tuned student model. 2) Generalization of the aligned teacher model.

| Models | PIQA | CSQA | ARC-E | ARC-C | GSM8K | MMLU | Average |
|--------|------|------|-------|-------|-------|------|---------|
| Vanilla Gemma-2B | 61.6 | 38.2 | 57.2 | 42.8 | 20.0 | 40.6 | 43.4 |
| + GPT-4-LLM | 60.2 | 42.8 | 55.8 | 46.2 | 18.6 | 39.4 | 43.8 |
| + Tulu-v2 | **65.2** | 42.8 | 62.2 | 49.6 | 18.8 | 40.6 | 46.5 |
| + UltraChat | 62.0 | 40.0 | 59.6 | 50.6 | 23.2 | **42.1** | 46.3 |
| + WizardLM | 63.6 | 41.4 | 56.4 | 49.2 | 26.2 | 40.5 | 46.2 |
| + OpenOrca | 64.6 | **43.6** | 61.0 | 48.4 | 23.6 | 40.0 | 46.9 |
| + ARTE | 63.4 | 42.9 | **63.6** | **57.2** | **32.0** | 41.4 | **50.1** |

Table 4: Accuracy (%) of the student model Gemma-2B fine-tuned with ARTE and academic baselines on out-of-domain reasoning benchmarks. Cells are colored blue if the method improves over Vanilla Gemma-2B, and orange if it declines.

## 4.1 GENERALIZATION OF THE FINE-TUNED STUDENT MODEL

To investigate the generalization ability of the fine-tuned student model Gemma-2B, we conduct experiments on the following academic reasoning benchmarks: 1) PIQA (Bisk et al., 2020) is a physical commonsense reasoning dataset that is designed to test the model's ability to build, craft, or manipulate objects using everyday physical knowledge. 2) CSQA (Talmor et al., 2019) is a question-answering benchmark targeting commonsense reasoning. 3) ARC-Easy and 4) ARC-Challenge (Clark et al., 2018) are multiple-choice question-answering benchmarks designed to test the model's ability to reason about scientific knowledge. 5) GSM8K (Cobbe et al., 2021) is a mathematical benchmark that is designed to test language models' ability in math and logic reasoning. 6) MMLU (Hendrycks et al., 2021) is a new benchmark designed to measure knowledge acquired during pretraining by evaluating models exclusively in zero-shot and few-shot settings.

Table 4 shows the results of our method and baselines on the six benchmarks. On average, our method outperforms the best baseline by 1.5% in accuracy. The results show that our method helps the student model Gemma-2B to achieve better generalization ability on reasoning tasks. In some tasks like PIQA, CSQA, and MMLU, the performance of our method is not the best but still competitive. While in more challenging tasks like ARC and GSM8K, which are both sampled from real-world grade school exams, our method outperforms all the baselines. This phenomenon reveals for more challenging tasks, more carefully curated examples are needed in knowledge distillation. Compared to the heuristic baselines, the tailored examples generated by the aligned teacher model are more effective in these tasks as they can better capture the student model's preferences.

## 4.2 GENERALIZATION OF THE ALIGNED TEACHER MODEL

Due to the high computational cost of preference collection and alignment, it is desirable that the aligned teacher model can generate tailored examples for unseen tasks and unseen student models.

**Generalization Across Tasks**  To investigate whether our aligned teacher model is capable of generating tailored examples for student models on unseen tasks, we conduct experiments on the PIQA, ARC-EASY, ARC-Challenge, and GSM8K benchmarks.

We repeat Step 1 *Knowledge Elicitation* to generate training examples on the four benchmarks. Specifically, we guide both original and aligned teacher models to generate 2500 training examples for each unseen benchmark. We train the student model Gemma-2B with the generated examples on the four unseen benchmarks separately with the same hyperparameters mentioned in Section 3.

| Models | PIQA | ARC-E | ARC-C | GSM8K | Average |
|--------|------|-------|-------|-------|---------|
| *zero-shot results* | | | | | |
| Original | 62.4 | 62.4 | 51.0 | 22.2 | 49.5 |
| Aligned | 65.6 | 59.4 | 51.2 | 25.6 | 50.5 |
| Delta | ↑ 3.2 | ↓ 3.0 | ↑ 0.2 | ↑ 3.4 | ↑ 1.0 |
| *three-shot results* | | | | | |
| Original | 65.0 | 64.2 | 51.8 | 23.4 | 51.1 |
| Aligned | 66.6 | 65.2 | 52.6 | 24.8 | 52.3 |
| Delta | ↑ 1.6 | ↑ 1.0 | ↑ 0.8 | ↑ 1.4 | ↑ 1.2 |

Table 5: Accuracy (%) of the student model Gemma-2B fine-tuned with training examples generated by the original teacher model and the aligned teacher model on out-of-domain reasoning benchmarks.

Results in Table 5 show that the student model fine-tuned with training examples generated by the aligned teacher model outperforms the student model fine-tuned with training examples generated by the original teacher model. This finding suggests that through preference alignment, the aligned teacher model gains a deeper understanding of the student model's preferences, which helps with unseen tasks. This enhanced understanding of the teacher model makes it easier to distill the specific abilities of the aligned teacher model to the student model compared to the original teacher model.

**Generalization Across Student Models**
To investigate the generalization ability of the aligned teacher model across unseen student models, we conduct experiments on the Big-Bench-Hard dataset. Specifically, we adopt Gemma-7B as the student model from the same family with a different capacity, Qwen1.5-1.8B and CodeGemma-2B as the student models with the same capacity but from different families or different domains. We train the student models with the training examples generated by the original teacher model and the aligned teacher model under the same hyperparameters mentioned in Section 3.

| Models | Gemma-7B | Qwen1.5-1.8B | CodeGemma-2B |
|---|---|---|---|
| *zero-shot results* | | | |
| Original | 49.0 | 36.3 | 38.7 |
| Aligned | 49.0 | 39.4 | 41.2 |
| Delta | 0.0 | ↑ 3.1 | ↑ 2.5 |
| *three-shot results* | | | |
| Original | 51.2 | 38.2 | 39.8 |
| Aligned | 51.0 | 40.3 | 42.7 |
| Delta | ↓ 0.2 | ↑ 2.1 | ↑ 2.9 |

Table 6: Accuracy (%) of the different student models fine-tuned with training examples generated by the original teacher model and the aligned teacher model on Big-Bench-Hard.

Table 6 shows the results of the student models trained with the original teacher model and the aligned teacher model, denoted as *Original* and *Aligned* respectively. The results show that in Qwen1.5-1.8B and CodeGemma-2B, the models from different families or different domains share similar parameter capacity with the Gemma-2B, and achieve better performance with the tailored examples generated by the aligned teacher model than the original teacher model. In Gemma-7B, the model from the same family with a larger parameter capacity does not present a promising improvement. This indicates that the language model with a similar parameter capacity shares similar preferences in training examples. This finding suggests that the aligned teacher model can be applied to different student models as long as they share similar parameter capacities.

## 5 INSIGHTS FOR GENERATING RATIONALE FOR LANGUAGE MODELS

In Step 2 *Preference Collection*, we collect the preference of small language models, such as Gemma-2B, towards the question-rationale pairs through one-shot in-context learning on the preference set. Through careful analysis of the collected preference scores, we conclude with two insights for generating a tailored question-rationale pair for language models with limited capacity:

**Insight 1: The more detailed the rationale does not necessarily mean the better the performance of the small language model.** In previous works (Hsieh et al., 2023; Mukherjee et al., 2023), the teacher model is guided to generate a detailed reasoning process with chain-of-thought (Wei et al., 2022; Yao et al., 2022) to provide more information for the small language model to imitate. However, after taking a closer look at the preference scores, we find that there is no significant linear correlation between the length of the rationale and the preference score, namely the accuracy of the small language model within one-shot in-context learning. Figure 2 shows the relationship between the word number of the rationale and the one-shot in-context learning accuracy of the small language model on boolean expressions and sports understanding tasks.

As shown in Figure 2, the rationale with most words does not necessarily mean the best performance of the small language model, while the rationale with moderate words (75-100 words in Boolean Expressions and 50-75 words in Sports Understanding) achieves better results. This indicates that *a rationale with complete and concise reasoning steps is more beneficial for the small language model to learn*, compared to a long meticulous rationale.

The reasons behind this are twofold: First, the *Lost-in-the-Middle Phenomenon* (Liu et al., 2024) of language models. Language models may lose information when the input context is too long.

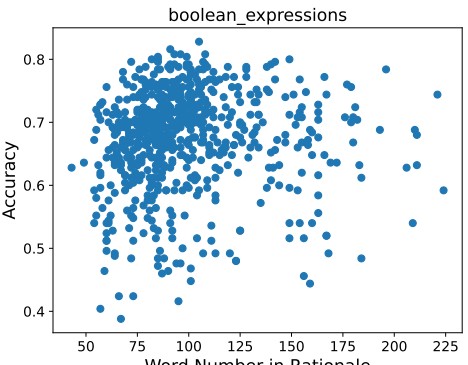 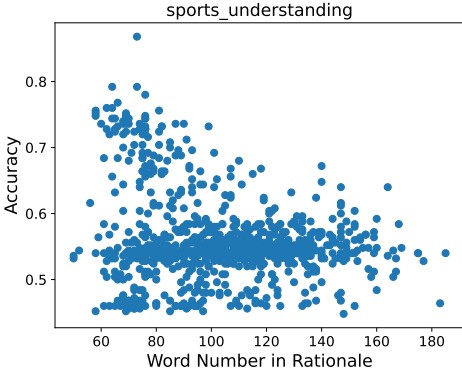

Figure 2: The relationship between the word number of the rationale and the one-shot in-context learning accuracy of the Gemma-2B on boolean expressions and sports understanding tasks.

For limited-capacity language models, when example rationales are too long, the small language model may get lost in the narrative and forget the question to solve. Second, the *Repetition Problem* (Welleck et al., 2019) of language models. In long meticulous rationales, the teacher model may repeat the same step multiple times, for example acting as a stack machine when solving a math problem. For limited-capacity language models, this repeated step may cause the small language model to get stuck in a loop and repeat the same step infinitely.

**Insight 2: Although different questions in the same task prefer different reasoning strategies, the small language model prefers consistent reasoning strategies for one task in Supervised Fine-tuning.** The diversity of tests in the training corpus is crucial for the pre-training stage of language models (Liu et al., 2019). To investigate the impact of the diversity of reasoning strategies in the training dataset, we fine-tuned the Gemma-2B with four different training datasets on Big-Bench-Hard. 1) *Randomly Selected*: the rationale of each question is randomly selected from different reasoning strategies. 2) *Most Preferred*: the rationale of each question is selected based on the highest preference score. 3) *Task Consistent*: all rationales are selected from within the same reasoning

| Method | BBH |
|---|---|
| Vanilla Gemma-2B | 35.57 |
| + Randomly Selected | 39.94 |
| + Most Preferred | 40.04 |
| + Task Consistent | 42.12 |
| + Aligned Teacher | 43.44 |

Table 7: Accuracy (%) of the Gemma-2B fine-tuned with different training examples on Big-Bench-Hard.

strategy for one task. 4) *Aligned Teacher*: the rationales are generated by the aligned teacher model with the small language model's preference. All the questions in these four training datasets are generated by the original teacher model to ensure fairness. Table 7 presents the performance of the small language model fine-tuned with four different training datasets on Big-Bench-Hard. The results present an interesting phenomenon: the performance of Gemma-2B fine-tuned with the *Most Preferred* dataset is similar to the *Randomly Selected* dataset, while *Task Consistent* and *Aligned Teacher* datasets both outperform the other two datasets by a decent margin. This indicates that *in the fine-tuning stage, especially when we try to enhance one or two specific capabilities of the small language model, for example, reasoning, the consistent reasoning strategies are more beneficial*. The reason behind this is that the small language model may get confused when the reasoning strategies are too diverse for one task, while the consistent reasoning strategies provide clear guidance for the small language model to imitate for specific capability enhancement.

## 6 RELATED WORK

**Instruction Tuning & Knowledge Distillation** Instruction tuning, which involves fine-tuning pre-trained language models on a labeled dataset known as an instruction-tuning dataset, is a widely adopted technique to enhance both task-specific performance and the ability to follow instructions. In this paper, we concentrate on improving the reasoning capabilities of lightweight language mod-

els through instruction tuning. Given the high costs and scalability challenges of creating human-labeled instruction-tuning datasets, Knowledge Distillation (KD) is frequently employed to distill such datasets from a powerful teacher model (Hinton et al., 2015). To enhance the quality of generated question-rationale pairs, Taori et al. (2023); Wang et al. (2022) suggested using ROUGE-L (Lin, 2004) to filter out repetitive questions, thereby increasing the diversity of the generated content. Xu et al. (2023) introduced Evo-Instruct, which incrementally rewrites seed questions into more complex ones, improving their sophistication. For generating rationales, Hsieh et al. (2023); Mukherjee et al. (2023) utilized chain-of-thought reasoning (Wei et al., 2022; Yao et al., 2022) to guide the teacher model in producing rationales that incorporate detailed reasoning steps, thus enhancing rationale quality. Additionally, Mitra et al. (2023) argued that the reasoning in rationales should be adaptable to different question types. While these approaches improve the quality of questions and rationales from the teacher model's perspective, they do not account for the student model's learning preferences. To address this limitation, we propose ARTE, a novel framework that aligns the teacher language model's outputs with the preferences of the student model, generating instruction-tuning datasets that are better tailored to the student model's needs.

**Alignment of Language Models**   Alignment of Language Models is about aligning the behavior with human or other models' preferences. It is initially proposed in the context of reinforcement learning from human feedback (Christiano et al., 2017), where the language model is trained to align with human preferences including safety, ethics, common sense, and other human values. Proximal Policy Optimization (PPO, Schulman et al., 2017) and Direct Preference Optimization (DPO, Rafailov et al., 2023) are two popular reinforcement learning algorithms used to achieve this alignment (Ouyang et al., 2022; Touvron et al., 2023; Meta, 2024; Yang et al., 2024). Since collecting human feedback is expensive and time-consuming, it has been extended to align the language model with preferences from other AI models or itself, known as RL from AI feedback (RLAF, Wang et al., 2023; Bai et al., 2022; Yuan et al., 2024). Our work is inspired by this line of research, where we align the teacher model with preference data collected from the student model to generate tailored training examples for reasoning tasks. Recently, some concurrent works have also inspired by RLAIF to improve the ability of the student model by adding an RLAIF-based loss term in the training process (Agarwal et al., 2024) or using DPO to replace the commonly used Supervised Fine-Tuning (Brown et al., 2020) (Zhang et al., 2024). These works focus on the training process, while our work focuses on the knowledge elicitation process, specifically on how to curate training data from LLMs.

## 7   CONCLUSION

In this study, we propose ARTE, a novel framework in Knowledge Distillation that aligns the teacher language model with the student language model's preferences to generate a tailored instruction tuning dataset, which is inspired by responsive teaching in pedagogy, which effectively lowers the learning barriers for students. Our framework consists of three main steps: *Knowledge Elicitation, Preference Collection*, and *Preference Alignment*. Extensive experiments on various academic benchmarks show that our method outperforms existing instruction-tuning datasets distilled from powerful LLMs by a large margin. Moreover, we investigate the generalization of the aligned teacher model, showing that the aligned teacher model can be generalized to other reasoning benchmarks and different student models.

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

APPENDIX

## A  LIMITATIONS

Although most of the data used in ARTE are automatically generated by the teacher model, there still needs some manual efforts to construct the prompts and collect the preference scores. Specifically, there are two main limitations in the current implementation of ARTE: First, in draft rationale generation, to elicit diverse and high-quality rationales, a set of carefully designed prompts is required. In this work, we hand-crafted the system prompts for each task using different reasoning techniques from the paper authors themselves. Totally, we constructed 4 prompts for each task as presented in Table 16. Recently, Wang & Zhou (2024) proposed to Chain-of-Thought Decoding (CoT-Decoding) to uncover reasoning processes of questions from language models without prompts. In the future, we will explore the possibility of using CoT-Decoding to generate a reasoning process automatically. Second, in preference collection, a set of labeled examples consisting of questions and answers is required to act as the validation set and preference set. Preference scores are collected on these labeled question-answer pairs to measure the preference of the student model towards the draft questions and rationales. In this work, we simply reuse the data from the original Big-Bench dataset as the validation set. In the future, we will explore the possibility of directly measuring the preference through the internal states of the student model (Kadavath et al., 2022).

## B  DISCUSSION

**Threshold of DPO data Selection**  When aligning the teacher model, we chose the top and bottom 25% to balance example quantity and quality. Higher thresholds like 5% may yield too few examples, while lower thresholds like 45% may include less discriminative ones. We further conducted an ablation study showing that 25% is a good balance point. We aligned teachers at different thresholds and used them to generate data for training the student model. The results of BBH are shown in Table 8:

| Threshold | 15% | 25% | 35% |
|---|---|---|---|
| **Student (0-shot)** | 39.7 | **42.0** | 41.6 |

Table 8: Different thresholds used to align the teacher model

**Scaling Relationship**  In this section, we discuss the feasibility of our approach for large-scale datasets and models.

We sampled 6,750 examples from each baseline dataset to conduct the experiment and ensure a fair comparison. To investigate the performance trend as data size varies, we trained the student model (Gemma-2B) with varying sizes of tailored data from the aligned teacher model (LLaMA-3-70B). The performance continued to improve with increasing data size, indicating promising scalability. The examples are shown in Table 9:

| Data Size | 1350 | 4050 | 6750 | 9450 | 12150 |
|---|---|---|---|---|---|
| **Student (0-shot)** | 39.2 | 40.3 | 42.0 | 45.2 | 46.0 |

Table 9: Different tailored data size used to train the student model

As for large-scale models, we would like to clarify that our approach is model-agnostic and can be applied to any LLMs. This is because our approach is based on In-Context Learning (Collect Student Preferences), Direct Preference Optimization (Align Teacher Model), and Supervised Fine-tuning (Train Student Model), which are general techniques available for all LLMs. Thus, our method is flexible and easily scalable to any of them.

**Human Evaluation**   Human evaluation is a good engineering effort to evaluate the model's performance. We manually check some cases of the answer between ARTE and others. Table 10 shows that with our ARTE distillation, the output is more specific and reaches a higher accuracy.

| Question | Original Teacher | ARTE |
|---|---|---|
| This SVG path element <path d=M̈ 5.41,37.88 L 1.69,19.06 M 1.69,19.06 L 0.30,43.85 M 0.30,43.85 L 5.41,37.88/̈>draws a Options: (A) circle (B) heptagon (C) hexagon (D) kite (E) line (F) octagon (G) pentagon (H) rectangle (I) sector (J) triangle | Let's break down the SVG path element step by step: 1. 'M 5.41,37.88': Move to the point (5.41, 37.88) 2. 'L 1.69,19.06': Draw a line to the point (1.69, 19.06) 3. 'M 1.69,19.06': Move to the point (1.69, 19.06) 4. 'L 0.30,43.85': Draw a line to the point (0.30, 43.85) 5. 'M 0.30,43.85': Move to the point (0.30, 43.85) 6. 'L 5.41,37.88': Draw a line back to the starting point (5.41, 37.88) From these steps, we can see that the shape has four sides, and all internal angles are right angles (90 degrees). This matches the definition of a... <ans>(H) rectangle</ans> | Let's break down the SVG path element step by step. The path starts at (5.41, 37.88) and moves to (1.69, 19.06), then to (0.30, 43.85), and finally back to (5.41, 37.88). This forms a shape with three sides, where all sides are of equal length, and opposite sides are parallel. The internal angles are also equal (120 degrees). This shape matches the definition of a <ans>(J) triangle</ans>. |
| Is the following sentence plausible? J̈ohn Carlson scored in the third period.̈ | Let's break it down! 1. John Carlson is a real NHL player, so that part is plausible. 2. The third period is a common term used in hockey to describe the final period of a game, so that's also plausible. 3. However, scoring is a term typically used in sports like soccer (or football), not hockey. In hockey, the score is usually kept by the goalie, not the players. Considering these points, I'd say the sentence is <ans>implausible</ans> | Let's break it down! * John Carlson is a real person and a professional ice hockey player. * Scoring is a common action in ice hockey, and the third period is a crucial period in a hockey game. * It's possible for a player to score in the third period. Considering these points, I conclude that the sentence is... <ans>plausible</ans> |

Table 10: Cases of the answer between ARTE and Original Teacher.

Besides, we follow the common practice in the field of Knowledge Distillation for Language Models, where human evaluation is not commonly conducted. For example, recent works like Phi-1(Gunasekar et al., 2023), Phi-2(Mic, 2023), Phi-3(Abdin et al., 2024), WizardLM(Xu et al., 2023), Orca(Mukherjee et al., 2023), Orca-2(Mitra et al., 2023), and SmoLM(Hug, 2024) did not conduct human evaluation and used similar benchmarks like we did to evaluate the model's performance, whose answers are explicit. Our approach aligns with these standards, ensuring consistency and comparability within the field.

# C   BROADER IMPACTS

The possible broader impacts of this work lie in the hallucination of language models. Language models have been shown to generate biased and harmful content (Bender et al., 2021; Zhang et al., 2023). In this work, we focus on the enhancement of the student model's specific capability, such as reasoning, which is not directly relevant to the generation of harmful content. It is worth noting that the user should be cautious when using the student model or the teacher model to generate text, especially when the generated text is used in critical applications such as medical diagnosis or legal advice.

## D  Big-Bench-Hard Category

We split the Big-Bench-Hard tasks into four categories based on the capabilities required by the tasks: (1) *logical reasoning*, (2) *commonsense reasoning*, (3) *math reasoning* and (4) *world knowledge*, denoted as *BBH-Logical*, *BBH-Commonsense*, *BBH-Math* and *BBH-Knowledge* respectively. Table 11 shows the detailed information of the tasks in Big-Bench-Hard.

| Category | Task | Quantity |
|----------|------|----------|
| BBH-Logic | dyck_languages, formal_fallacies_syllogisms_negation, logical_deduction, navigate, object_counting, penguins_in_a_table, temporal_sequences, tracking_shuffled_objects, web_of_lies, word_sorting | 10 |
| BBH-CS | causal_judgment, date_understanding, disambiguation_qa, hyperbaton, reasoning_about_colored_objects, ruin_names, snarks | 7 |
| BBH-Math | boolean_expressions, geometric_shapes, multistep_arithmetic | 3 |
| BBH-Knowl | movie_recommendation, salient_translation_error_detection, sports_understanding | 3 |

Table 11: Big-Bench-Hard task categories.

## E  Repeating Knowledge Elicitation to Generate Training Examples

After the *Preference Alignment* step, we need to repeat the *Knowledge Elicitation* step to generate training examples for the student model with the aligned teacher model. Specifically, first, we prompt the aligned teacher model with question generation prompts as shown in Table 15 to generate draft examples. We will decode the question generation prompts with decoding temperature 1.0 multiple times until we get enough draft questions. Then, for each question, we prompt the aligned teacher model with naive *Step-by-Step* prompts as shown in Table 12 to generate rationales with decoding temperature 0.0.

Task Description: {Task Description}
Question: {Question}
Answer: Let's think step by step. {to be completed by the teacher model}

Table 12:  Naive *Step-by-Step* prompt for the teacher model.

## F  Direct Preference Optimization Details

In the *Preference Alignment* step, we align the teacher model with the student model's preferences through Direct Preference Optimization (DPO) (Rafailov et al., 2023). Here we provide the details of constructing the DPO dataset

**DPO dataset for Question Generation**    For each task, we construct the DPO dataset for question generation using the following template. The question generation prompt is constructed based on the task description and seed questions, detailed in Table 15. The preferred response is randomly selected from the draft questions with top-25% preference scores, while the dispreferred response is randomly selected from the draft questions with bottom-25% preference scores.

**DPO dataset for Rationale Generation**    For each draft question, we construct the DPO dataset for rationale generation using the following template, The naive *Step-by-Step* prompt is constructed

| Input Prompt $x$: | { The Question Generation Prompt } |
|---|---|
| Preferred Response $y_w$: | {Preferred Question} |
| Dispreferred Response $y_l$: | {Dispreferred Question} |

Table 13: DPO dataset template for question generation.

| Input Prompt $x$: | { The Naive *Step-by-Step* Prompt } |
|---|---|
| Preferred Response $y_w$: | {Preferred Rationale} |
| Dispreferred Response $y_l$: | {Dispreferred Rationale} |

Table 14: DPO dataset template for rationale generation.

based on the task description and the draft question, detailed in Table 12. The idea of using naive *Step-by-Step* prompts is inspired by *Prompt Erasure* (Mitra et al., 2023), which lets the teacher model learn to select the most suitable reasoning strategy to generate tailored training examples for the student model. The preferred response is the draft rationale to the draft question with the highest preference scores, while the dispreferred response is the draft rationale with the lowest preference scores.

Finally, we mix the DPO dataset for question generation and rationale generation together to train the teacher model with DPO.

## G  QUESTION GENERATION PROMPT

We provide some prompts for guiding the teacher model to generate questions in the **Knowledge Elicitation** step.

| Task | Prompt |
|---|---|
| boolean_expressions | boolean expressions is a task whose description is as follows: Evaluate the result of a random Boolean expression.
I will provide you 3 example questions about this task in the following. Based on the following example, I want you to generate 5 more questions about boolean expressions with brainstorming.
You may not stick to context, but the question should be related to boolean expressions.
The output should be like this:
<Question_x><input>...</input><target>...</target></Question_x>
x can be any number, but it should be unique.

<Question_1><input>not ( ( not not True ) ) is</input><target>False</target></Question_1>
<Question_2><input>True and False and not True and True is</input><target>False</target></Question_2>
<Question_3><input>not not ( not ( False ) ) is</input><target>True</target></Question_3> |
| causal_judgment | causal judgment is a task whose description is as follows: Answer questions about causal attribution.
I will provide you 3 example questions about this task in the following. Based on the following example, I want you to generate 5 more questions about causal judgment with brainstorming.
You may not stick to context, but the question should be related to causal judgment.
The output should be like this:
<Question_x><input>...</input><target>...</target></Question_x>
x can be any number, but it should be unique.

<Question_1><input>How would a typical person answer each of the following questions about causation? |

Frank T., had an ongoing dispute with his neighbor over a stretch of land and one day decided to shoot his neighbor in the body. Frank T. had no experience with guns, his hand slipped on the barrel of the gun, and the shot went wild. Nonetheless, the bullet bounced off a large boulder several feet away and hit the neighbor's body, causing significant injury. Did Frank T. intentionally shoot his neighbor in the body?
Options:
- Yes
- No</input><target>No</target></Question_1>
<Question_2><input>How would a typical person answer each of the following questions about causation?
Suzy and Billy are working on a project that is very important for our nation's security. The boss tells them both: "Be sure that you are here at exactly 9 am. It is absolutely essential that you arrive at that time." Both Billy and Suzy arrive at 9 am. As it happens, there was a motion detector installed in the room where they arrived. The motion detector was set up to be triggered if at least one person appeared in the room at the same time. So the motion detector went off. Did Billy cause the motion detector to go off?
Options:
- Yes
- No</input><target>Yes</target></Question_2>
<Question_3><input>How would a typical person answer each of the following questions about causation?
George and his sister Lena reunite at their parent's house for Thanksgiving. Whereas George just got into medical school, Lena is unhappy in her marriage and recently lost her job. Over the course of the day, George and Lena get into a number of heated arguments. Later in the afternoon, they play a game of darts. They split the first two games, and the third game was close until the end. Who will win comes down to George's last shot. If he hits a high point region, he wins; if he hits a low point region, Lena wins. George thinks of the difficult time Lena is having, and he really wants to let her win. He aims the dart at the low point region. He sets up his shot and the dart lands in the low point region. After his shot, Lena wins the game and is very happy. Did George hit the low point region intentionally?
Options:
- Yes
- No</input><target>Yes</target></Question_3>

| | |
|---|---|
| date_understanding | date understanding is a task whose description is as follows: Infer the date from context.
I will provide you 3 example questions about this task in the following. Based on the following example, I want you to generate 5 more questions about date understanding with brainstorming.
You may not stick to context, but the question should be related to date understanding.
The output should be like this:
<Question_x><input>...</input><target>...</target></Question_x>
x can be any number, but it should be unique.

<Question_1><input>Today is Christmas Eve of 1937. What is the date 10 days ago in MM/DD/YYYY?
Options:
(A) 12/14/2026
(B) 12/14/1950
(C) 12/14/2007
(D) 12/14/1937
(E) 07/14/1938
(F) 12/14/1988</input><target>(D)</target></Question_1>
<Question_2><input>Tomorrow is 11/12/2019. What is the date one year ago from today in MM/DD/YYYY?
Options:
(A) 09/04/2018
(B) 11/11/2018
(C) 08/25/2018
(D) 11/02/2018 |

(E) 11/04/2018</input><target>(B)</target></Question_2>
<Question_3><input>Jane and John married on Jan 2, 1958. It is their 5-year anniversary today. What is the date tomorrow in MM/DD/YYYY?
Options:
(A) 01/11/1961
(B) 01/03/1963
(C) 01/18/1961
(D) 10/14/1960
(E) 01/03/1982
(F) 12/03/1960</input><target>(B)</target></Question_3>

| | | |
|---|---|---|
| movie_recommendation | movie recommendation is a task whose description is as follows: Recommend movies similar to the given list of movies. | |

movie_recommendation

movie recommendation is a task whose description is as follows: Recommend movies similar to the given list of movies.

I will provide you 3 example questions about this task in the following. Based on the following example, I want you to generate 5 more questions about movie recommendations by brainstorming.

You may not stick to context, but the question should be related to movie recommendations.

The output should be like this:

<Question_x><input>...</input><target>...</target></Question_x>

x can be any number, but it should be unique.

<Question_1><input>Find a movie similar to Star Wars Episode IV - A New Hope, Indiana Jones and the Last Crusade, Star Wars Episode V - The Empire Strikes Back, The Big Lebowski:
Options:
(A) Tetsuo
(B) the Ironman
(C) The Princess Bride
(D) The Barkley Marathons The Race That Eats Its Young
(E) Bug</input><target>(C)</target></Question_1>
<Question_2><input>Find a movie similar to Twister, The Silence of the Lambs, Independence Day, Braveheart:
Options:
(A) They Shoot Horses
(B) Don't They
(C) Forrest Gump
(D) The Salton Sea
(E) Extreme Days</input><target>(C)</target></Question_2>
<Question_3><input>Find a movie similar to Minority Report, Total Recall, Inside Out, Forrest Gump:
Options:
(A) Phenomena
(B) Lilting
(C) Catwoman
(D) Edge of Tomorrow</input><target>(D)</target></Question_3>

Table 15: Prompts for question generation.

# H   RATIONALE GENERATION PROMPT FOR DIFFERENT REASONING TECHNIQUES

We provide some prompts for guiding the teacher model to generate rationale with different reasoning techniques in the **Knowledge Elicitation** step.

| Task | Prompt |
|------|--------|
| boolean_expressions | Image you are an expert in Boolean expression evaluation. Now you will be given a random Boolean expression, you should first evaluate the expressions inside brackets, then follow the order of operations from highest priority to lowest priority namely "not", "and", "or", respectively, and finally evaluate the result of the random Boolean expression. Remember you should output your final answer in the end like <ans>True</ans>or <ans>False</ans> |
| | You are an expert in Math. Given a random Boolean expression, you should first recall the rules of Boolean algebra and then evaluate the expression step by step. Finally, you should provide the result of the expression. Remember you should output your final answer in the end like <ans>True</ans>or <ans>False</ans> |
| | Evaluate the result of a random Boolean expression. Remember you should output your final answer in the end like <ans>True</ans>or <ans>False</ans> |
| | Given you are a binary classification question, solve the question step by step as follows: 1. Read the question and options 2. Find the best option among the remaining ones. Remember you should output your final answer in the end like <ans>True</ans>or <ans>False</ans> |
| causal_judgment | I want you to act as a judge in a causal judgment process. There are 3 key points you should consider when assessing whether an action was intentional or not. 1. Intentionality and Outcome: When evaluating an action, consider the actor's intentions and the sequence of events leading to the outcome. If the actor clearly intended the outcome and acted to bring it about, the action is intentional. 2. Accidents and Unintended Consequences: If the outcome resulted from an accident or slip, and not from a deliberate act aimed at causing that specific outcome, it is considered unintentional. Unexpected or uncontrollable factors can influence the final result, diverging from the actor's original intent. 3. Contribution to Outcome: In cases where multiple factors contribute to an outcome, analyze each factor's role in producing the final effect. If an agent's actions directly contribute to triggering an event, they can be considered a cause, even if other factors are also involved. Carefully assess the situation and the actor's mindset to make a fair and accurate judgment. Based the three key points, given one scenario, you should first analyze the situation and the actor's mindset. Then, decide which key point is the most relevant to the scenario. Finally, you should conclude by answering the question of whether an action was intentional or not. Remember you should output your final answer in the end like <ans>True</ans>or <ans>False</ans> |
| | You are given a scenario and asked to figure out whether the person in the scenario intentionally caused the outcome. Use your common sense and the information given in the scenario to answer the question step by step. Remember you should output your final answer in the end like <ans>True</ans>or <ans>False</ans> |
| | Answer questions about causal attribution. Remember you should output your final answer in the end like <ans>True</ans>or <ans>False</ans> |
| | Given you are a binary classification question, solve the question step by step as follows: 1. Read the question and options 2. Find the best option among the remaining ones. Remember you should output your final answer in the end like <ans>True</ans>or <ans>False</ans> |

| | | |
|---|---|---|
| date_understanding | | You are given a known date and asked to calculate a date based on a specific time interval or event. To solve these types of questions, follow these steps: 1. Identify the known date provided in the question. 2. Understand the time interval or event mentioned (e.g., "10 days ago", "one year ago", "5-year anniversary"). 3. Calculate the required date by counting backward or forwards from the known date according to the interval or event. 4. Compare the calculated date with the options given to find the correct answer. Finally, select the option that matches the calculated date. Remember you should output your final answer in the end like <ans>(A)</ans>or <ans>(B)</ans>, etc. |
| | | Use your knowledge of the calendar and common sense. Carefully read the question and the options. And then, handle the math calculation step by step to find the answer. Finally, select the correct answer.Remember you should output your final answer in the end like <ans>(A)</ans>or <ans>(B)</ans>, etc. |
| | | Infer the date from context. Remember you should output your final answer in the end like <ans>(A)</ans>or <ans>(B)</ans>, etc. |
| | | Given a multi-choice question, your task is to solve the question step by step as follows: 1. Read the question and options. 2. Eliminate the options that are clearly wrong. 3. Find the best option among the remaining ones. Remember you should output your final answer in the end like <ans>(A)</ans>or <ans>(B)</ans>, etc. |
| movie_recommendation | | Image you are a movie buff and you are asked to recommend movies similar to a given list of movies. Follow the steps below to find the answer. (1) Identify the common genres and themes of the movies in the list. (2) Consider the time period of production, focusing on classics or movies from a specific era. (3) Compare the options provided, looking for films that match the identified genres, themes, and production period. (4) Select the option that best aligns with the criteria established from the given list of movies. Remember you should output your final answer in the end like <ans>(A)</ans>or <ans>(B)</ans>, etc. |
| | | Image you are a movie buff and you are asked to recommend movies similar to a given list of movies. You need to think about the genre, the year of production, the actors, and the overall style of the movies. Think step by step and eliminate the options that are not similar to the given list of movies. Choose the option that is most similar to the given list of movies. Remember you should output your final answer in the end like <ans>(A)</ans>or <ans>(B)</ans>, etc. |
| | | Recommend movies similar to the given list of movies. Remember you should output your final answer in the end like <ans>(A)</ans>or <ans>(B)</ans>, etc. |
| | | Given a multi-choice question, your task is to solve the question step by step as follows: 1. Read the question and options. 2. Eliminate the options that are clearly wrong. 3. Find the best option among the remaining ones. Remember you should output your final answer in the end like <ans>(A)</ans>or <ans>(B)</ans>, etc. |

Table 16: Task prompts for different reasoning techniques.

# I ONE-SHOT IN-CONTEXT LEARNING TEMPLATE FOR PREFERENCE COLLECTION

We provide the in-context learning prompt for the student model in the **Preference Collection** step.

---

Task Description: {Task Description}
Remember you should include your final answer with the tag `<ans>` and `</ans>`.
Question: {Example Question}
Answer: Let's think step by step. {Example Rationale}
Question: {Test Question}
Answer: Let's think step by step. {to be completed by the student model}

---

Table 17: One-shot in-context learning prompt for preference collection.

# J LICENSES FOR EXISTING ASSETS

The names of the licenses for each asset used in this paper are detailed below.

| Asset | License |
|---|---|
| BBH | MIT License |
| GPT-4-LLM | Apache License Version 2.0 |
| Tulu-v2 | ODC-BY |
| WizardLM | Apache License Version 2.0 |
| OpenOrca | MIT License |
| UltraChat | MIT License |
| PIQA | Academic Free License v3.0 |
| CSQA | MIT License |
| ARC-E | CC-BY 4.0 |
| ARC-C | CC-BY 4.0 |
| GSM8K | MIT License |
| MMLU | MIT License |
| Gemma-2B | Gemma Terms of Use |
| Gemma-7B | Gemma Terms of Use |
| Qwen1.5-1.8B | Tongyi Qianwen RESEARCH LICENSE |
| CodeGemma-2B | Gemma Terms of Use |
| Llama-3-70B-Instruction | META LLAMA 3 COMMUNITY LICENSE |

Table 18: Licenses for each asset in the paper.

