# OpenReview forum: "Aligning Teacher with Student Preferences for Tailored Instruction Tuning Dataset Generation"
_ICLR.cc/2025/Conference — ICLR 2025 Conference Withdrawn Submission_

### Official Review · Reviewer_Gj7R · 2024-11-03

**Soundness:** 2
**Presentation:** 3
**Contribution:** 2
**Rating:** 5
**Confidence:** 4

**Summary:**

The paper presents the ARTE framework, designed to enhance the effectiveness of instruction tuning for lightweight language models (LMs) on reasoning tasks. Traditional instruction-tuning approaches often prioritize the quality of questions and rationales based on the teacher model's perspective, overlooking the preferences and learning styles of the student model. ARTE addresses this by aligning the outputs of the teacher model with the student’s learning preferences, using a process inspired by responsive teaching from pedagogy. ARTE's contributions include significant improvements in reasoning benchmarks by tailoring instruction-tuning datasets to student preferences, showing gains across logical, commonsense, math, and knowledge reasoning tasks.

**Strengths:**

1. The thought process of the authors is novel, as I see it from the perspective of AGI, where instruction-rationale datasets/tasks can be constructed using the interaction of LLMs when the LLMs reach or surpass human-level performance

2. The authors provide a thorough empirical analysis of the approach across several LLMs and reasoning benchmarks.

3. The authors provide insightful findings on rationale length and consistency. They observe that longer rationales do not always correlate with better performance in lightweight models and that consistency in reasoning style within tasks is more beneficial than varied strategies.

**Weaknesses:**

1. ICL score is not the right fit to evaluate the quality of the generated instruction.
2. The writing style of the methodology section is not formal, as in, experimental details are provided in the methods section, rather than using notations and providing a generic write-up about the methodology.
3. A theoretical insight of the given approach is missing.
4. Lack of reproducibility, code and datasets are not released along with the paper.

**Questions:**

1. Can you explain why the ICL score of the student model directory correlates with the quality of the generation question? ICL score being less may mean the rationale was also incorrect, so how does this separation occur, as in, for instruction-rationale pairs, how does a single ICL score indicate good instruction and good rationale?

2. Was PEFT used to train the teacher model? No information is provided since as per my knowledge, full fine-tuning using DPO of LLaMA3-80B is not possible using 8 80GB A100 GPUs.

3. I see a reduction in delta i.e., improvement on general benchmarks as the number of shots increases. Does delta further decrease when num shots increase from 3 to 5 or more? One claim could be that ARTE significantly performs better when num shots are less if this is the case.

4. **Suggestion: Please write the method in a more generic way, as the paper presents a framework, rather than some engineering solution to a particular task. Please do not mention the dataset used in the method section.

**Details Of Ethics Concerns:**

No ethics review is needed as per my understanding.

---

### Official Review · Reviewer_PXVg · 2024-11-03

**Soundness:** 1
**Presentation:** 3
**Contribution:** 2
**Rating:** 3
**Confidence:** 4

**Summary:**

This paper proposes a technique for leveraging larger-capacity language models (teacher models) to synthesize more downstream-task-effective instruction-tuning datasets for small-capacity models (student models). The authors achieve this goal by (1) first using the original teacher model to generate more question-rationale pairs; (2) evaluating the quality of each data pair based on the student's performance; (3) aligning the teacher model to generate higher-quality data pairs using DPO; and (4) using the aligned teacher to generate the final instruction-tuning dataset. The experiments are primarily conducted on Big-Bench-Hard and other reasoning benchmarks, which demonstrate that their method can indeed improve the quality of the teacher's generations.

**Strengths:**

* The authors propose a new knowledge-distillation paradigm to improve both the quality of the teacher-generated data and the student's performance. The results are promising, and the aligned teacher holds the potential to generate high-quality data for other tasks and students that were not seen or used during alignment.
* The paper is well-written and easy to understand.

**Weaknesses:**

* I am not satisfied with the main experimental settings in Section 3. The authors compare the quality of their generated data with other baselines, such as OpenOrca and UltraChat. However, most of these baseline datasets are designed for **general instruction-following ability**, resulting in a clear distribution shift between the baseline training data and the BBH benchmarks. In contrast, the training data utilized by ARTE is tailored to the evaluation tasks based on both the seed dataset for generation and the validation set for DPO data selection, thereby reducing the training-test gap. In summary, I do not think the experimental results in Tables 3 and 4 come from a "fair comparison."
* Most of the experiments are conducted on 2B-sized student models and the Llama3-70B-Instruct as the teacher model, which casts doubt on whether this method can be applied to more diverse and realistic application settings. For example: (1) Can the teacher model effectively learn the preferences of 7B or even larger student models? (2) What is the minimum size of the teacher model required for effectively learning the student's preferences?

**Questions:**

1. Regarding the experimental results presented in Tables 3 and 4, I believe the authors should compare the **methods** used by these baselines to construct their datasets rather than comparing the datasets themselves. Reproducing their dataset construction methods using the same seed dataset would more convincingly demonstrate the strengths of ARTE, rather than directly utilizing the existing data.
2. Can the entire ARTE framework be applied to larger-sized student models or smaller-sized teacher models?
3. The authors claim that longer rationales in training data do not necessarily lead to better student performance due to the *Lost-in-the-Middle Phenomenon* or the *Repetition Problem*. However, I did not find any experimental results or case studies in the paper to support this claim.
4. The ablation study in App. B only examines the scaling of the student's training dataset size. Could the authors also perform an ablation on the size of the teacher's DPO training dataset to assess its impact on the final generation data quality?
5. Similar works [1,2] show that their data synthesis methods can be performed iteratively to generally boost the final generation quality. Can ARTE also benefit from performing multiple rounds of data generation, data evaluation (preference collection), and teacher alignment?

---
[1] Self-Play Fine-Tuning Converts Weak Language Models to Strong Language Models
[2] Self-Instruct: Aligning Language Models with Self-Generated Instructions

---

### Official Review · Reviewer_AXRL · 2024-11-04

**Soundness:** 2
**Presentation:** 2
**Contribution:** 2
**Rating:** 5
**Confidence:** 4

**Summary:**

The paper proposes ARTE (Aligning TeacheR with StudenT PreferencEs), a framework designed to improve the reasoning performance of lightweight language models (LMs) by aligning the data generation process with student model preferences. Unlike traditional instruction-tuning methods that solely optimize for teacher model outputs, ARTE incorporates student preferences to create tailored question-rational pairs. This alignment involves generating initial outputs from the teacher model, gathering feedback from the student on these outputs, and using Direct Preference Optimization to refine the teacher model's responses to match student needs. Experiments on academic reasoning benchmarks show that ARTE-based datasets significantly enhance student performance across various reasoning tasks compared to conventional instruction-tuning datasets, suggesting that student-aligned data generation can boost both in-domain performance and generalization across tasks and models​.

**Strengths:**

1. This paper is clearly written and easy to follow.
2. The experiments on reasoning tasks are comprehensive.
3. The content of this paper is sufficient.

**Weaknesses:**

1. The novelty and contribution of this paper should be further justified as some of the related papers are not included and discussed:

[1] One-Shot Learning as Instruction Data Prospector for Large Language Models. (ACL24)
[1] proposed a method that utilizes one-shot in-context learning with a validation set to find the data that suits the LLMs to be trained, which looks similar to your step2 method “The one-shot in-context learning performance of the student model”.

[2] Selective Reflection-Tuning: Student-Selected Data Recycling for LLM Instruction-Tuning. (ACL24 findings)
The motivation of [2] is “Many recent methods focus on improving the data quality but often overlook the compatibility of the data with the student model being finetuned”, which seems related to yours “Current methods for generating these instruction-tuning datasets tend to focus solely on the quality of the questions and rationales from the teacher model’s perspective, often neglecting the learning preferences of the student language model.”

[3] LLM2LLM: Boosting LLMs with Novel Iterative Data Enhancement. (ACL24 findings)
[3] also proposed a teacher-student interactive pipeline that “uses a teacher LLM to enhance a small seed dataset by augmenting additional data that can be used for fine-tuning on a specific task”. It also seems to try to find the data that student LLMs are not good at, which is related to your work.

I understand the difficulty of doing a thorough literature review, but I think the inclusion and discussions of these papers are highly prefered.

2. As far as I know, GPT-4-LLM, Tulu-v2, UltraChat, and WizardLM datasets are originally built for general instruction-following abilities and they do not explicitly contain the reasoning task training data, while your method seems to use the training data. This might lead to potential unfair comparisons. For example, in Table 3, Vanilla Gemma-2B + previous datasets settings lead to a really low performance compared with your ablations. Maybe a general instruction-following benchmark like Alpaca_Eval is required if you want to compare your method with the previous ones.

3. Related to the previous point, I think Insight 1 should be further justified. More analysis should be made on why “the more detailed the rationale does not necessarily mean the better the performance”. Is it possible that it is only because the evaluation metric is not suitable to the long and diverse responses?

4. For both Insight 1 and 2, I think you might need to constrain your conclusion into “finetuning on sub-task”, because these conclusions might not be true for the general instruction-following abilities, especially for this benchmark:
[4] Evaluating Large Language Models at Evaluating Instruction Following. (ICLR24)

**Questions:**

N/A

---

### Official Review · Reviewer_ujVi · 2024-11-04

**Soundness:** 2
**Presentation:** 3
**Contribution:** 2
**Rating:** 3
**Confidence:** 4

**Summary:**

This paper studies instruction-tuning data curation for reasoning tasks. It deploys a teacher model and a student model to collaboratively generate data via three steps: knowledge elicitation, preference collection, and preference alignment. In experiments, the authors validate the proposed method on Big-Bench-Hard and investigate the generalization of the aligned teacher model.

**Strengths:**

1. This paper presents an instruction-tuning data curation method for reasoning tasks.

2. The main results show the proposed method can outperform baselines by a large margin.

**Weaknesses:**

1. The motivation drawing from 'responsive teaching' in pedagogy lacks sufficient depth and clarity. While there are conceptual connections to pedagogical principles, the paper needs stronger theoretical justification for why the proposed data curation pipeline is effective. For instance, the authors should either demonstrate how their method provides a more on-policy approach compared to baselines, or present theoretical analysis that supports their method design.

2. In the main experiment, only one teacher model and one student model are used. Testing one or two more settings (using alternative teacher-student model combinations) would strengthen the results.

3. The training overhead and time efficiency are not discussed.

4. The baselines are limited. Only instruction-tuning datasets are taken as baselines. However, there are also multiple KD methods or alignment methods that could serve as baselines.

5. Typos:
Line 162: trained -> train, for consistency, using the present tense in the method section would be appropriate.
Line 209: to target tasks -> on target tasks.
Line 229: Since We compare -> We compare.
Line 512: have also inspired -> have also been inspired.

**Questions:**

See above.

---

### Note · Authors · 2024-11-27

I have read and agree with the venue's withdrawal policy on behalf of myself and my co-authors.